# REWARD-FOCUSED FINE-TUNING OF POCKET-AWARE DIFFUSION MODELS VIA DIRECT PREFERENCE OPTIMIZATION

## ABSTRACT

Diffusion models have recently been found promising for Structure-based drug design (SBDD). Yet, how to effectively fine-tune the models for generating more desirable drug-like molecules given the relatively scarce pocket-ligand data remains challenging. With the recent success of aligning diffusion models with preference data, we introduce **R**eward-**F**ocused **F**ine-**T**uning (**RFFT**) which is a novel framework for fine-tuning pretrained pocket-aware diffusion models using direct preference optimization (DPO). Using a reward score and self-generated ligand pairs from the pretrained model, RFFT constructs data with winner-loser pairs as feedback and fine-tunes the model with DPO accordingly. The process can be repeated iteratively to gain continuous improvement. To illustrate its effectiveness, we apply RFFT to fine-tune a diffusion model TargetDiff recently proposed for SBDD. Our empirical results demonstrate that TargetDiff-RFFT after fine-tuning can gain substantial improvement on generation quality. Also, its performance is highly competitive to the existing state-of-the-art baselines, being first place in Chemical Property analysis and second place in Binding Affinity analysis. Surprisingly, our substructure analysis results show that RFFT not only preserves but actually enhances the model's fidelity to real data distributions.

## 1 INTRODUCTION

Structure-based drug design (SBDD) is a fundamental problem in therapeutic design and biological discovery. In the past few years, diffusion-based models, following their success in image generation (Rombach et al., 2022; Dai et al., 2023; Podell et al., 2023), have been actively explored for SBDD. Given pairwise pocket-ligand datasets, diffusion models can be trained to generate drug-like molecules as ligands (Guan et al., 2023a; Lin et al., 2025a; Schneuing et al., 2024; Lin et al., 2024; Guan et al., 2023b). The success of learning powerful diffusion models relies on the availability of high quality pocket-ligand datasets, which however are scarce.

Reinforcement learning based on human feedback (RLHF) has been demonstrated to be effective in fine-tuning large language models (Achiam et al., 2023; Touvron et al., 2023) and diffusion models (Rombach et al., 2022; Dai et al., 2023; Podell et al., 2023) leveraging only human preference data, or via an implicit reward model (Rafailov et al., 2023). While the RLHF approach may suffer from issues such as convergence difficulties, model drift, and inefficiency (Xiao et al., 2024; Wang et al., 2024), direct preference optimization (DPO) (Rafailov et al., 2023) was recently proposed to reformulate the RLHF problem with a simple classification loss via reparameterization. Diffusion-DPO (Wallace et al., 2024) learns diffusion models using DPO with an implicit reward function introduced to favor generation of more desirable data. Yet, even with these recent developments, applying the approach to SBDD is still challenging, primarily due to the scarcity of preference data of ligands where expert knowledge on drugs and medicines are required for preparing them.

Alternative approaches have also been explored to enhance diffusion generation quality. For example, domain knowledge (e.g., chemical substructure) can be incorporated into diffusion models to improve the validity of the generated ligands (Lin et al., 2024; Guan et al., 2023b). This however often compromises accuracy in docking pose prediction and explicit prioritization of desirable molecules (Song et al., 2024), together with reduced sampling efficiency. In addition, classifier-guided diffusion generation (Dhariwal & Nichol, 2021; Ma et al., 2024) has been shown effective in

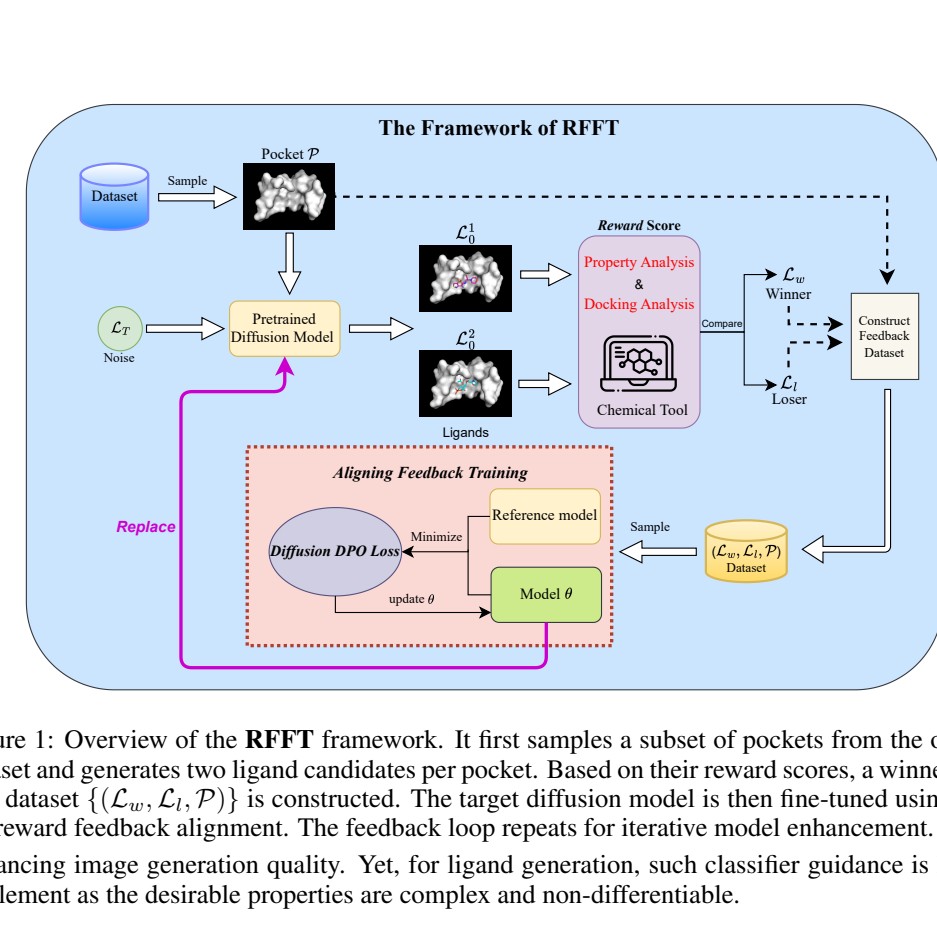

Figure 1: Overview of the **RFFT** framework. It first samples a subset of pockets from the original dataset and generates two ligand candidates per pocket. Based on their reward scores, a winner-loser pair dataset $\{(\mathcal{L}_w, \mathcal{L}_l, \mathcal{P})\}$ is constructed. The target diffusion model is then fine-tuned using DPO for reward feedback alignment. The feedback loop repeats for iterative model enhancement.

enhancing image generation quality. Yet, for ligand generation, such classifier guidance is hard to implement as the desirable properties are complex and non-differentiable.

To this end, we propose **R**eward-**F**ocused **F**ine-**T**uning (**RFFT**) which is a novel learning framework making use of self-generated samples and a reward score derived from desirable drug-like properties of a ligand molecule as feedback for fine-tuning a diffusion model. RFFT samples pairs of ligands from a target diffusion model and computes their reward scores to obtain a set of winner-loser ligand pairs as feedback. The DPO framework is then applied for fine-tuning the target model. As the model after fine-tuning will generate samples of higher quality, we repeat the feedback loop by sampling and fine-tuning the model iteratively so that it can be further refined to have a higher chance of generating ligands of even higher quality. In other words, this iterative process allows **RFFT** to systematically identify the model's shortcomings and reinforces its strengths, thereby enhancing the overall ligand generation quality.

To differentiate from a recently proposed model ALIDIFF-E$^2$PO (Gu et al., 2024) which also adopts DPO for fine-tuning diffusion models but from the preference pairs sampled from the pocket-ligand dataset, RFFT samples from the target model which makes the iterative model enhancement via the feedback loop possible. Furthermore, sampling ligands from the target model to be paired allows RFFT to have more control over the feedback design. Sampling from the pocket-ligand data as adopted by ALIDIFF-E$^2$PO will unavoidably pair up ligands of different sizes, which will bias the model training as molecular size and binding affinity are known to be highly correlated (Lin et al., 2025b).

The contributions of this paper can be summarized as:

- We design a novel learning framework based on direct preference optimization (DPO) for fine-tuning diffusion models to generate pocket-specific ligands, where self-generated samples and a reward score are used to form a feedback-loop for iterative model enhancement.
- We demonstrate that diffusion models like TargetDiff fine-tuned using RFFT based on a very limited number of self-generated samples can outperform the case using a larger sample selected from the original dataset within the DPO framework.
- Experimental results demonstrate that **RFFT** can achieve state-of-the-art performance in regarding different metrics such as molecular validity, chemical properties, and binding affinity, with the substructure fidelity of the target diffusion model also improved, validating the rationality of our design.

## 2 RELATED WORK

**Alignment of Diffusion Models with Preference Feedback**    Aligning large language models (LLMs) with human preference feedback for fine-tuning has been extensively explored. Reinforcement Learning from Human Feedback (RLHF) is one of the key underlying approaches for optimizing LLM alignment with human expectation, demonstrating remarkable effectiveness (Ouyang et al., 2022; Touvron et al., 2023). While RLHF is also known for its sensitivity to hyperparameters, vulnerability to local optima, training instability, and high demand on computational resources and data, Direct Preference Optimization (DPO) (Rafailov et al., 2023) is one of the recently proposed method for making the feedback alignment computationally more viable. While LLM alignment has received considerable attention, alignment of diffusion models remains is relatively underexplored. Some initial efforts focus on fine-tuning diffusion models using higher-quality image datasets (Podell et al., 2023; Rombach et al., 2022; Dai et al., 2023), while some researchers leveraged datasets with enhanced text accuracy (Betker et al., 2023). Also, guided models have been employed in DOODL (Wallace et al., 2023), DRAFT (Clark et al., 2024), and AlignProp (Prabhudesai et al., 2023) to improve inference capabilities. They so far mostly focus on image generation and generalizing them to molecule generation is non-trivial. Reinforcement learning-based diffusion models have demonstrated promising alignment capabilities, albeit primarily under simple prompt conditions. Alternatively, recent advances in diffusion models have leveraged implicit reward models trained on pairwise preference data to achieve more efficient alignment (Wallace et al., 2024; Yang et al., 2024). Nevertheless, these methods rely heavily on the availability of human-annotated preference data, which constrains their applicability and robustness.

**Generative Models for SBDD**    Generative modeling has rapidly advanced SBDD. Earlier voxel- and grid-based methods such as LIGAN (Ragoza et al., 2022) and 3DSBDD (Luo et al., 2021) suffer from limited 3D invariance and grid resolution issues. With EGNNs (Satorras et al., 2021), models like Pocket2Mol (Peng et al., 2022) and GraphBP (Liu et al., 2022) can achieve direct atomic coordinate generation, which are limited by the autoregressive sampling.

Diffusion-based approaches, such as DiffBP (Lin et al., 2025a), TargetDiff (Guan et al., 2023a), and DiffSBDD (Schneuing et al., 2024), have established themselves as state-of-the-art methods for one-shot molecular generation conditioned on protein pockets. Recent works have further enhanced these models by integrating domain knowledge. For example, D3FG (Lin et al., 2024) and DecompDiff (Guan et al., 2023b) decompose molecules into functional groups or scaffolds to more effectively guide the generative process. IPDiff (Huang et al., 2024) leverages prior knowledge of protein-ligand interactions to improve molecular generation, while MolCraft (Qu et al., 2024) utilizes a framework of Bayesian Flow Network (BFN) (Graves et al., 2023) to learn hyperparameters of the data distribution to enable better hybrid attribute modeling and accelerate SBDD molecule generation.

Despite these advances, most of the existing methods primarily focus on modeling pocket-ligand structures and often lack explicit mechanisms to control molecular properties or binding affinities. Notably, Gu et al. (2024), whose work is most closely related to ours, directly selects winner-loser pairs from the original dataset for learning the alignment. This approach, while effective, mainly utilizes the preferences inherent in the original data for the model fine-tuning, lacking mechanisms to provide more focused feedback to address the model's deficiencies. Furthermore, it tends to overlook important correlations between molecular size, chemical properties, and binding affinity. To overcome these limitations, our proposed RFFT framework learns from preference pairs self-generated using the target diffusion model which is to be fine-tuned, thereby guiding the generative process towards molecules with desirable properties and improved docking scores. This approach aims to facilitate the discovery of drug-like molecules with both favorable chemical characteristics and enhanced binding affinity.

## 3 PRELIMINARIES

### 3.1 DIFFUSION MODELS

**Diffusion Models for Continuous and Discrete Data**    Diffusion models generate data by gradually corrupting samples from the data distribution $q(\mathbf{x}_0)$ (or $q(\mathbf{c}_0)$ for discrete data) through a forward process, and then learning to reverse this process via denoising steps. In the continuous

case, noise is added according to a predefined schedule, transforming the data into a tractable prior (Ho et al., 2020; Song et al., 2021). The reverse process is modeled as a parameterized Gaussian:

$$p_\theta(\mathbf{x}_{t-1}|\mathbf{x}_t) = \mathcal{N}\left(\mathbf{x}_{t-1}; \boldsymbol{\mu}_\theta(\mathbf{x}_t, t), \frac{\sigma_{t|t-1}^2 \sigma_{t-1}^2}{\sigma_t^2}\mathbf{I}\right). \tag{1}$$

The training objective is to minimize the Evidence Lower Bound (ELBO), which reduces to a sum of KL divergences at each timestep:

$$\sum_{t=2}^{T} D_{\mathrm{KL}}\left(q(\mathbf{x}_{t-1} \mid \mathbf{x}_t, \mathbf{x}_0) \,\|\, p_\theta(\mathbf{x}_{t-1}|\mathbf{x}_t)\right). \tag{2}$$

Specifically, the single-step KL term $L_t$ can be equivalently written in either mean or noise parameterization, weighted by functions of the signal-to-noise ratio (Rombach et al., 2022; Kingma et al., 2021; Ho et al., 2020):

$$L_t = \mathbb{E}_{\mathbf{x}_0, \boldsymbol{\epsilon}, t}\left[\omega_1(\lambda_t)\left|\boldsymbol{\mu}_t(\mathbf{x}_t, \mathbf{x}_0) - \boldsymbol{\mu}_\theta(\mathbf{x}_t, t)\right|^2\right] = \mathbb{E}_{\mathbf{x}_0, \boldsymbol{\epsilon}, t}\left[\omega_2(\lambda_t)\left|\boldsymbol{\epsilon}_t - \boldsymbol{\epsilon}_\theta(\mathbf{x}_t, t)\right|^2\right], \tag{3}$$

where $\boldsymbol{\epsilon} \sim \mathcal{N}(0, \mathbf{I})$ and $t \sim \mathcal{U}(0, T)$. The noisy sample at timestep $t$ is drawn from $q(\mathbf{x}_t|\mathbf{x}_0) = \mathcal{N}\left(\mathbf{x}_t; \alpha_t \mathbf{x}_0, \sigma_t^2 \mathbf{I}\right)$, and the signal-to-noise ratio is $\lambda_t = \alpha_t^2/\sigma_t^2$.

For discrete data, multinomial diffusion models extend this framework to categorical variables, representing each data point as a one-hot vector and adding uniform noise at each step (Hoogeboom et al., 2021; Guan et al., 2023a). The forward process is defined as:

$$q(\mathbf{c}_t|\mathbf{c}_{t-1}) = \mathcal{C}\left(\mathbf{c}_t; (1 - \beta_t)\mathbf{c}_{t-1} + \frac{\beta_t}{K}\mathbf{1}\right), \tag{4}$$

with the marginal distribution after $t$ steps given by

$$q(\mathbf{c}_t|\mathbf{c}_0) = \mathcal{C}\left(\mathbf{c}_t; \bar{\alpha}_t \mathbf{c}_0 + \frac{1 - \bar{\alpha}_t}{K}\mathbf{1}\right), \tag{5}$$

where $\bar{\alpha}_t = \prod_{\tau=1}^{t}(1 - \beta_\tau)$.

The training objective similarly involves minimizing the KL divergence between the true and model posteriors at each step:

$$\sum_{k=1}^{K} \boldsymbol{\theta}_{\mathrm{post}}(\mathbf{c}_t, \mathbf{c}_0)_k \cdot \log \frac{\boldsymbol{\theta}_{\mathrm{post}}(\mathbf{c}_t, \mathbf{c}_0)_k}{\boldsymbol{\theta}_{\mathrm{post}}(\mathbf{c}_t, \hat{\mathbf{c}}_0)_k}. \tag{6}$$

where $\mathbf{c}_0$ and $\hat{\mathbf{c}}_0$ represent the real initial state and the predicted initial state. In both cases, the core principle is to learn a parameterized reverse process by minimizing the KL divergence between the true and model posteriors at each timestep, enabling the generation of high-fidelity samples from complex data distributions.

## 3.2 DIFFUSION-DPO FOR STRUCTURE-BASED DRUG DESIGN

In structure-based drug design (SBDD), the goal is to generate ligand molecules that optimally bind to a given protein pocket. We represent a binding system as a protein-ligand pair $(\mathcal{P}, \mathcal{L})$, where $\mathcal{P} = (\boldsymbol{V}_{\mathrm{rec}}, \boldsymbol{E}_{\mathrm{rec}})$ denotes the protein receptor (nodes: atoms or residues; edges: interaction bonds), and $\mathcal{L} = (\boldsymbol{V}_{\mathrm{lig}}, \boldsymbol{E}_{\mathrm{lig}})$ denotes the ligand (nodes: atoms; edges: bonds). The generative model aims to learn the conditional probability $p(\mathcal{L} \mid \mathcal{P})$ for de novo molecule generation, i.e., filling the protein pocket with ligand atoms.

To incorporate human or oracle preferences, we adopt the Bradley-Terry (BT) model, which formulates the probability that ligand $\mathcal{L}_0^w$ is preferred over $\mathcal{L}_0^l$ given protein pocket $\mathcal{P}$ as:

$$p_{\mathrm{BT}}(\mathcal{L}_0^w \succ \mathcal{L}_0^l \mid \mathcal{P}) = \sigma(r(\mathcal{P}, \mathcal{L}_0^w) - r(\mathcal{P}, \mathcal{L}_0^l)) \tag{7}$$

where $\sigma$ is the sigmoid function, and $r(\mathcal{P}, \mathcal{L}_0)$ is a reward function parameterized by a neural network $\phi$, estimated via binary classification:

$$L_{\mathrm{BT}}(\phi) = -\mathbb{E}_{\mathcal{P}, \mathcal{L}_0^w, \mathcal{L}_0^l}\left[\log \sigma(r_\phi(\mathcal{P}, \mathcal{L}_0^w) - r_\phi(\mathcal{P}, \mathcal{L}_0^l))\right] \tag{8}$$

To directly optimize the generative model with respect to such preferences, we employ Direct Preference Optimization (DPO) in the diffusion model framework (Diffusion-DPO) (Rafailov et al., 2023; Wallace et al., 2024). The objective for SBDD can be formulated as:

$$\max_{p_\theta} \mathbb{E}_{\mathcal{P},\mathcal{L}_{0:T} \sim p_\theta(\mathcal{L}_{0:T}|\mathcal{P})} \left[ r(\mathcal{P}, \mathcal{L}_0) \right] - \beta D_{\mathrm{KL}} \left( p_\theta(\mathcal{L}_{0:T} \mid \mathcal{P}) \| p_{\mathrm{ref}}(\mathcal{L}_{0:T} \mid \mathcal{P}) \right), \tag{9}$$

where $r(\mathcal{P}, \mathcal{L}_0)$ is the marginal reward for the generated ligand, and $p_{\mathrm{ref}}$ is a reference diffusion model.

By applying an upper bound and an approximation, the loss function for Diffusion-DPO in SBDD is:

$$L_{\text{DPO-Diffusion}}(\theta) = -\mathbb{E}_{(\mathcal{L}_0^w, \mathcal{L}_0^l), t, \mathcal{L}_{t-1:t}^{w,l}} \log \sigma \left( \beta T \log \frac{p_\theta(\mathcal{L}_{t-1}^w | \mathcal{L}_t^w, \mathcal{P})}{p_{\mathrm{ref}}(\mathcal{L}_{t-1}^w | \mathcal{L}_t^w, \mathcal{P})} - \beta T \log \frac{p_\theta(\mathcal{L}_{t-1}^l | \mathcal{L}_t^l, \mathcal{P})}{p_{\mathrm{ref}}(\mathcal{L}_{t-1}^l | \mathcal{L}_t^l, \mathcal{P})} \right) \tag{10}$$

where the preference label is $\mathcal{L}_0^w \succ \mathcal{L}_0^l$.

# 4 METHODOLOGY

In this section, we present RFFT, a framework that fine-tunes pocket-conditioned diffusion models using self-generated preference pairs. Our approach consists of three key components: (1) constructing size-matched preference ligand pairs by sampling from a target diffusion model, (2) formulating a DPO-based objective for fine-tuning the model, and (3) implementing fine-tuning driven by feedback iteratively for continuous model improvement.

## 4.1 PREFERENCE PAIR DATASET CONSTRUCTION

For fine-tuning a target diffusion model, rather than sampling ligand pairs from existing databases like CrossDocked2020 (Francoeur et al., 2020), we sample candidates from the model itself, which forms a closed-loop feedback process. This self-sampling approach enables the model to be self-guided for addressing its own weakness in ligand generation via the DPO framework.

To ensure fair subsequent optimization, we construct preference pairs with identical molecular size (same heavy-atom count). The size control is important for the following two reasons: (i) Chemical metrics are size-dependent. E.g., QED and SA decrease with molecular size (Bickerton et al., 2012; Cremer et al., 2024), while binding affinity increases with heavy-atom count up to a threshold (Kuntz et al., 1999). Ligands with size differences inevitably confound property comparison; (ii) Size imbalances introduce gradient variance F.

Following the standard protocols (Ragoza et al., 2022; Guan et al., 2023a; Lin et al., 2025b), we randomly select 3,000 training pockets and generate two ligand candidates of the same size per pocket. After filtering invalid molecules, approximately 2,600 valid pairs remain. For each pair, we compute a composite reward score which combines scores of the chemical properties and the docking affinity:

$$R = 0.1 \times (\text{QED} + \text{SA} + \text{Lipinski}) + 0.7 \times (-\text{Vina score}) \tag{11}$$

All scores are normalized (see Appendix G). For each ligand pair generated based on the pocket $\mathcal{P}$, the one with a higher reward score is the winner $\mathcal{L}^w$, and the other is the loser $\mathcal{L}^l$, forming a preference tuple $(\mathcal{L}^w, \mathcal{L}^l, \mathcal{P})$. We subsample 80% (2,096) of these tuples for training.

## 4.2 DPO OBJECTIVE FOR FINE-TUNING DIFFUSION MODELS

We fine-tune the pocket-conditioned diffusion model using the DPO framework. The reverse process is factorized into continuous coordinates and discrete atom types (Guan et al., 2023a):

$$p_\theta(\mathcal{L}_{t-1} \mid \mathcal{L}_t, \mathcal{P}) = p_\theta^1(\mathbf{x}_{t-1} \mid \mathbf{x}_t, \mathcal{P}) \, p_\theta^2(v_{t-1} \mid v_t, \mathcal{P}), \tag{12}$$

Given a preference pair $\mathcal{L}_0^w \succ \mathcal{L}_0^l$ and reference model $p_{\mathrm{ref}}$, the Diffusion-DPO loss is:

$$L_{\text{DPO}}(\theta) = -\mathbb{E}_{t, \mathcal{L}_{t-1:t}^{w,l}} \log \sigma \left( \beta T \log \frac{p_\theta(\mathcal{L}_{t-1}^w \mid \mathcal{L}_t^w, \mathcal{P})}{p_{\mathrm{ref}}(\mathcal{L}_{t-1}^w \mid \mathcal{L}_t^w, \mathcal{P})} - \beta T \log \frac{p_\theta(\mathcal{L}_{t-1}^l \mid \mathcal{L}_t^l, \mathcal{P})}{p_{\mathrm{ref}}(\mathcal{L}_{t-1}^l \mid \mathcal{L}_t^l, \mathcal{P})} \right). \tag{13}$$

Rewriting as KL divergence differences yields:

$$L(\theta) = -\mathbb{E}\big[\log \sigma(-\beta T \cdot \Delta)\big], \qquad \Delta = \lambda_1(t)\Delta_\epsilon + \lambda_2(t)\Delta_\pi. \tag{14}$$

The continuous component $\Delta_\epsilon$ measures noise-prediction improvements:

$$\Delta_\epsilon = \big(\|\epsilon^w - \epsilon_\theta(\mathcal{L}_t^w, t)\|_2^2 - \|\epsilon^w - \epsilon_{\text{ref}}(\mathcal{L}_t^w, t)\|_2^2\big) - \big(\|\epsilon^l - \epsilon_\theta(\mathcal{L}_t^l, t)\|_2^2 - \|\epsilon^l - \epsilon_{\text{ref}}(\mathcal{L}_t^l, t)\|_2^2\big), \tag{15}$$

while the discrete component $\Delta_\pi$ measures categorical alignment:

$$\begin{aligned}
\Delta_\pi = &\sum_{k=1}^{K}\bigg[\pi_\theta(\mathcal{L}_t^w, t)_k \log \frac{\pi_\theta(\mathcal{L}_t^w, t)_k}{\pi(\mathbf{c}_t^w, \hat{\mathbf{c}}_0^w)_k} - \pi_{\text{ref}}(\mathcal{L}_t^w, t)_k \log \frac{\pi_{\text{ref}}(\mathcal{L}_t^w, t)_k}{\pi(\mathbf{c}_t^w, \hat{\mathbf{c}}_0^w)_k}\bigg] \\
&- \sum_{k=1}^{K}\bigg[\pi_\theta(\mathcal{L}_t^l, t)_k \log \frac{\pi_\theta(\mathcal{L}_t^l, t)_k}{\pi(\mathbf{c}_t^l, \hat{\mathbf{c}}_0^l)_k} - \pi_{\text{ref}}(\mathcal{L}_t^l, t)_k \log \frac{\pi_{\text{ref}}(\mathcal{L}_t^l, t)_k}{\pi(\mathbf{c}_t^l, \hat{\mathbf{c}}_0^l)_k}\bigg].
\end{aligned} \tag{16}$$

Using standard forward noising $\mathbf{x}_t = \alpha_t \mathbf{x}_0 + \sigma_t \epsilon$ with $\epsilon \sim \mathcal{N}(0, I)$, and constant timestep weights $\lambda_1, \lambda_2$ (Ho et al., 2020; Kingma et al., 2021), this objective increases the likelihood of winners relative to losers while staying close to $p_{\text{ref}}$. Full derivations are in Appendix B.

### 4.3 ITERATIVE FINE-TUNING DRIVEN BY FEEDBACK

RFFT implements an iterative fine-tuning process driven by feedback that creates a closed-loop learning system. Given a target pretrained diffusion model $p_{\theta^{(0)}}$, RFFT performs multiple rounds of fine-tuning to systemically enhance the model performance using self-generated preference pairs as feedback (Fig. 1).

In each round $r \geq 1$, RFFT will (1) sample training pockets $\mathcal{P}_r$ and generates two ligand candidates of the same size per pocket using $p_{\theta^{(r-1)}}$, (2) filter invalid molecules and compute reward scores using Eq. (11), (3) assign winners/losers to form preference tuples $(\mathcal{L}^w, \mathcal{L}^l, \mathcal{P}_r)$, (4) fine-tune the target diffusion model by optimizing the DPO objective in Eq. (10), keeping a frozen reference model $p_{\text{ref}}$ to prevent drift, and (5) update $p_{\theta^{(r)}}$ for the next round while keeping $p_{\text{ref}}$ fixed.

## 5 EXPERIMENTS

We used the publicly released pretrained TargetDiff model from CBGBench and fine-tuned it with **RFFT**, yielding the model denoted as **TargetDiff-RFFT**. We conduct 4 rounds of model fine-tuning through DPO. For each round, we randomly sample 3,000 pockets and generate pairs of ligand candidates (same size) per pocket. After filtering out invalid or disconnected molecules, we then randomly select 80% of these winner-loser pairs for training. The hyper-parameter details of the experimental settings are provided in Appendix H.

**Baselines** For performance evaluation, we adopt the following baselines: autoregressive models (AR): 3DSBDD (Luo et al., 2021), Pocket2Mol (Peng et al., 2022), and GraphBP (Liu et al., 2022); diffusion models (DM): DiffSBDD (Schneuing et al., 2024), DiffBP (Lin et al., 2025a), and TargetDiff (Guan et al., 2023a); and Bayesian flow networks model (BFN): MolCraft (Qu et al., 2024). For methods that released neither model weights nor test-time samples in CBGBench (Lin et al., 2025b), we report the performance reported by CBGBench (marked with an asterisk). We implemented an IPDiff variant (Huang et al., 2024) within CBGBench, re-

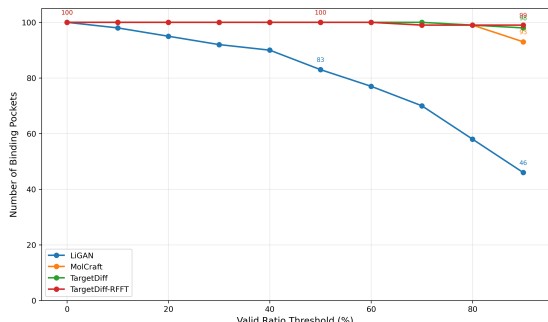

Figure 2: Pocket Count by Valid Ratio Threshold

taining the TargetDiff backbone while incorporating a pretrained interaction module. For fairness, we strictly follow CBGBench experimental settings for the original baselines, including the number of training iterations.

Note that baselines using CNN-based models (Ragoza et al., 2022; Pinheiro et al., 2024) are not included in our detailed comparison due to their limited performance on generating valid ligands as

Table 1: Performance Comparison on Chemical Properties and Interaction Analysis

| Type | Methods | Validity% | QED | SA | LPSK | Avg. Rank | Vina Score | | Vina Min | | Vina Dock | | | | Avg. Rank |
|---|---|---|---|---|---|---|---|---|---|---|---|---|---|---|---|
| | | | | | | | $E_{vina}$ | IMP% | $E_{vina}$ | IMP% | $E_{vina}$ | IMP% | MPBG% | LBE | |
| AR | 3DSBDD* | 54.00 | 0.48 | 0.63 | 4.72 | 4 | - | 3.99 | -3.75 | 17.98 | -6.45 | 31.46 | **9.18** | 0.3839 | 5.5 |
| | GraphBP* | 66.00 | 0.44 | 0.64 | **4.73** | 4.25 | - | 0.00 | - | 1.67 | -4.57 | 10.86 | -30.03 | 0.3200 | 8.25 |
| | Pocket2Mol | 39.32 | 0.375 | 0.649 | 4.63 | 5.75 | -5.13 | 26.02 | -5.89 | 39.82 | -6.82 | 45.13 | -3.51 | **0.4545** | 4.75 |
| DM | DiffSBDD | 40.53 | 0.391 | 0.614 | 4.60 | 6.5 | - | 1.07 | - | 13.54 | -5.86 | 31.84 | -14.48 | 0.3024 | 7.625 |
| | DiffBP* | 78.00 | 0.47 | 0.59 | 4.47 | 6.25 | - | 8.60 | - | 19.68 | -7.34 | 49.24 | 6.23 | 0.3481 | 5.75 |
| | IPDiff | 93.22 | 0.465 | 0.576 | 4.35 | 7 | -4.32 | 36.85 | -5.79 | 45.84 | -7.38 | 51.86 | 5.1 | 0.3592 | 4.625 |
| | TargetDiff | 97.05 | 0.487 | 0.598 | 4.57 | 4.25 | -5.73 | 38.51 | -6.44 | 47.14 | -7.39 | 52.17 | 5.53 | 0.3545 | 3.625 |
| | **TargetDiff-RFFT** | **97.41** | **0.500** | 0.606 | 4.61 | **3** | **-6.54** | 49.55 | **-6.99** | 54.65 | -7.70 | 58.68 | 8.87 | 0.3608 | 1.875 |
| BFN | MolCraft* | 95.00 | 0.48 | **0.66** | 4.39 | 3.75 | -6.15 | **54.25** | **-6.99** | **56.43** | **-7.79** | 56.22 | 8.38 | 0.3638 | **1.75** |

they are not 3D invariant (as mentioned in Section 2). As shown in Fig. 2, LIGAN (Ragoza et al., 2022) generates samples of ligands with low validity across many pockets, indicating limited generalization to unseen pockets. To focus on the effectiveness of using feedback for enhancement diffusion generation quality, models requiring prior knowledge on chemical substructures (like functional groups) are also not included (e.g., D3FG (Lin et al., 2024), DecompDiff (Guan et al., 2023b)).

## 5.1 EXPERIMENTAL RESULTS

We conducted evaluation on 200 sampled ligands across the pockets in the test set against the baselines. Our evaluation metrics focus on several key aspects of generation quality: chemical properties, and protein-ligand interaction. In addition, we also analyze the substructures between the generated and reference molecules.

**Performance on Chemical Properties** We evaluated the generated molecules based on several key chemical properties, including validity ($\uparrow$), QED ($\uparrow$), SA ($\uparrow$), and Lipinski (LPSK, $\uparrow$) scores. Validity is defined following the CBGBench protocol, where a molecule is considered valid if its largest fragment contains more than 85% of the total atoms after 3D reconstruction—using either the 'Refine' method or Open Babel (O'Boyle et al., 2011) tool. As reported in Table 1, TargetDiff-RFFT attains the best performance on Validity and QED. Although its SA is lower than some autoregressive (AR) baselines and MolCraft, and its LPSK lags behind certain AR models, it nevertheless achieves the best average performance (3) across chemical-property metrics. Notably, as QED, SA, and Lipinski form the components of the reward score as feedback, the consistent increases of these scores of TargetDiff-RFFT as compared to TargetDiff indicate effective alignment.

**Performance on Interaction** For pocket-ligand interaction assessment, we follow CBGBench and report three AutoDock Vina protocols (Trott & Olson, 2010; Eberhardt et al., 2021): (1) Score Only (direct scoring), (2) Minimize (energy minimization before scoring), and (3) Dock (full docking process). We evaluate mean Vina energy ($E_{vina}$, $\downarrow$) and improvement percentage (IMP%, $\uparrow$) for all three protocols, and mean percent binding gap (MPBG, $\uparrow$) and ligand binding efficacy (LBE, $\uparrow$) for Dock. As shown in Table 1, TargetDiff-RFFT is highly competitive across the affinity metrics. Specifically, for $E_{vina}$ it ranks first in Score Only, ties for first in Minimize, and places second in Dock (slightly behind MolCraft); for IMP it ranks second in Score Only and Minimize, and first in Dock. It further attains the second-best MPBG while LBE is worse than some AR models, yielding an overall average rank of 1.875, slightly behind MolCraft (1.75). Importantly, incorporating feedback learning consistently improves over the original TargetDiff across all metrics, with substantial gains.

**Performance on Substructure Analysis** To assess our model's ability to capture the underlying data distribution after incorporating feedback preference learning, we employ Jensen-Shannon divergence (JSD, $\downarrow$) and mean absolute error (MAE, $\downarrow$) I to quantify the differences between generated and reference distributions of atom types, ring types, and functional groups (Lu et al., 2021). As shown in Table 2, we surprisingly find that TargetDiff-RFFT significantly outperforms the original TargetDiff across all evaluation dimensions. For atom types, our model achieves the lowest MAE; for ring types, it secures the best JSD and second-best MAE; for functional groups, it attains the second best JSD and best MAE. This improvement indicates that feedback learning enables TargetDiff to fine-tune itself to generate molecules which can better align with the real drug molecule distributions, validating the rationality of our design. Overall, our model ties best in average performance with the SOTA model MolCraft.

Table 2: Performance Comparison on Substructure Analysis

| Type | Metrics / Methods | Atom type | | Ring type | | Functional Group | | Avg. Rank |
|------|---------|-----------|---|-----------|---|------------------|---|------|
| | | $JSD_{at}$ | $MAE_{at}$ | $JSD_{rt}$ | $MAE_{rt}$ | $JSD_{fg}$ | $MAE_{fg}$ | |
| AR | 3DSBDD* | 0.086 | 0.844 | 0.319 | 0.246 | 0.268 | 0.049 | 4.67 |
| | GraphBP* | 0.164 | 1.227 | 0.506 | 0.438 | 0.626 | 0.071 | 8.33 |
| | Pocket2Mol | 0.085 | 1.422 | 0.392 | 0.391 | 0.356 | 0.066 | 6.33 |
| DM | DiffSBDD | 0.053 | 0.632 | 0.385 | 0.344 | 0.552 | 0.071 | 5.83 |
| | DiffBP* | 0.259 | 1.549 | 0.453 | 0.407 | 0.535 | 0.067 | 8 |
| | IPDiff | 0.116 | 0.624 | 0.286 | 0.203 | 0.395 | 0.057 | 5 |
| | TargetDiff | 0.053 | 0.247 | 0.230 | 0.157 | 0.291 | 0.045 | 2.5 |
| | **TargetDiff-RFFT** | 0.054 | **0.219** | **0.219** | 0.135 | 0.269 | **0.041** | **2** |
| BFN | MolCraft* | **0.049** | 0.321 | 0.247 | **0.026** | **0.120** | 0.048 | **2** |

In summary, across the three evaluation aspects, TargetDiff-RFFT exhibits consistent and robust improvements. *Chemical Property*: It achieves the best validity and improves QED/SA/Lipinski over TargetDiff. It gives the best average chemical-property ranking (3.0), though SA and Lipinski are not the highest among all baselines. *Interaction*: Across Vina-based protocols, it remains in top-2 - being first in mean energy and second in IMP under Score Only; tied first in energy and second in IMP under Minimize; second in energy, first in IMP, and second-best in MPBG under Dock, while LBE lags behind the best AR and BFN baselines. Overall, our model is second-best in affinity analysis, slightly behind MolCraft, and significantly outperforms TargetDiff across all interaction metrics. *Substructure Analysis*: Our empirical results show that RFFT via the feedback alignment can enhance the distributional fidelity. It achieves best atom-type MAE; best JSD and second-best MAE for ring types; and competitive JSD with best MAE for functional groups, yielding an overall average that ties MolCraft for best. Taken together, TargetDiff-RFFT delivers strong and stable performance across protocols, achieving top-tier chemical properties and consistently top-2 results in binding affinity and substructure fidelity, highlighting the effectiveness of our RFFT approach in generating drug-like, pocket-compatible molecules.

We further verify that the aforementioned gains persist across backbone architectures, hyperparameter choices, and reward variants (including Boltz-2; Eq. 51). See Appendix Table 5 for more details.

## 5.2 ITERATIVE IMPROVEMENT OF TARGETDIFF-RFFT

To illustrate how well the performance of TargetDiff is enhanced over iterations, we recorded the model performance after each round of iterative training. We also conduct experiments to compare the cases between fine-tuning on model-sampled winner-loser pairs (adopted by RFFT) and fine-tuning on 12,546 pairs directly extracted from the original dataset. As shown in Table 3, after just one round of training, RFFT achieves performance comparable to applying DPO directly on the original dataset which is much larger than the self-generated ones used by RFFT. Across most metrics, TargetDiff-RFFT shows continuous improvement and has not reached a performance plateau after four rounds of fine-tuning. Notably, the attributes considered in the reward score consistently exhibit upward trends throughout the iterative process. This demonstrates the power of the iterative feedback in enabling more efficient learning.

Table 3: Performance comparison of TargetDiff fine-tuned under different settings. TargetDiff-RFFT improves consistently over iterative rounds of fine-tuning and outperforms TargetDiff fine-tuned using a much larger dataset of preference pairs.

| Method | QED | SA | LPSK | Validity% | Vina Score | | Vina Min | | Vina Dock | | | |
|--------|-----|-----|------|-----------|------------|---|----------|---|-----------|---|---|---|
| | | | | | $E_{vina}$ | IMP% | $E_{vina}$ | IMP% | $E_{vina}$ | IMP% | MPBG% | LBE |
| TargetDiff | 0.487 | 0.598 | 4.57 | 97.05 | -5.73 | 38.51 | -6.44 | 47.14 | -7.39 | 52.17 | 5.53 | 0.3545 |
| DPO on original dataset | 0.460 | 0.622 | 4.58 | 94.08 | -6.05 | 42.39 | -6.67 | 48.53 | -7.33 | 53.46 | 5.01 | 0.3524 |
| 1-Round RFFT | 0.491 | 0.598 | 4.58 | 96.98 | -6.05 | 43.19 | -6.63 | 50 | -7.49 | 54.61 | 6.18 | 0.358 |
| 2-Round RFFT | 0.494 | 0.6 | 4.58 | 96.97 | -6.23 | 44.7 | -6.73 | 51.91 | -7.54 | 55.90 | 5.71 | 0.3585 |
| 3-Round RFFT | 0.494 | 0.599 | 4.59 | 97.06 | -6.44 | 48.08 | -6.89 | 53.26 | -7.70 | 56.37 | 9.70 | 0.3642 |
| 4-Round RFFT | 0.500 | 0.606 | 4.61 | 97.41 | -6.54 | 49.55 | -6.99 | 54.65 | -7.70 | 58.68 | 8.87 | 0.3608 |

## 5.3 CASE STUDY

In Figure 3, we visualize four ligand pairs generated under the same pocket condition from the held-out test set (pocket "4yhj"). Both methods were initialized from identical noisy ligands with the same random seed and sampling settings for fairness. The displayed pairs were randomly selected

without cherry-picking. In case (1), both methods produce structurally similar ligands, but Target-Diff generates a seven-membered ring while TargetDiff-RFFT yields a six-membered ring, which is easier to synthesize due to lower ring strain. In case (2), TargetDiff-RFFT generates a simpler scaffold with fewer rings, improving synthetic accessibility despite a slight QED decrease. In cases (3) and (4), TargetDiff-RFFT avoids complex ring systems, achieving better overall QED and SA scores. We provide more case study examples in Appendix L.

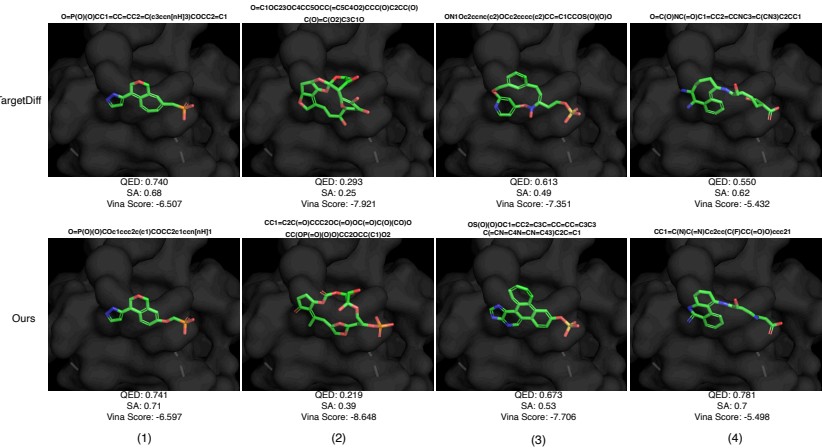

Figure 3: Four randomly selected ligand pairs from the randomly chosen test pocket "4yhj". For each pair, TargetDiff and TargetDiff-RFFT are sampled with identical noise seeds and inference settings.

## 6 CONCLUSION AND LIMITATIONS

In this work, we present **Reward-Focused Fine-Tuning** (RFFT), a framework that strengthens diffusion-based SBDD by learning from preference pairs (winner-loser) generated by the model itself. By explicitly applying self-generated direct preference optimization (DPO), RFFT optimizes its strengths while recognizing its shortcomings. Extensive experiments show consistent gains across molecular validity, chemical properties, pocket–ligand binding affinity, and distributional fidelity. Iterative feedback training over four rounds yields sustained improvements and enables RFFT to surpass state-of-the-art baselines. Notably, TargetDiff-RFFT after four rounds is one of only two methods that consistently rank within the top three across all key evaluation axes. Moreover, TargetDiff-RFFT yields the highest number of pockets meeting each success-rate threshold. Despite relying on a much smaller, model-generated preference set, RFFT matches methods trained on substantially larger winner-loser collections sampled from the original dataset. Our empirical results further indicate that iterative feedback training clearly outperforms the setting with DPO directly applied to the original dataset.

While our results are promising, the current framework assumes a linear mapping from reward signals to molecular properties, which may under-represent nonlinear structure–activity relations. Future extensions include: (i) multi-reward/multi-objective formulations, (ii) online preference learning from model committees and human feedback, (iii) learnable reward functions to better capture complex trade-offs, and (iv) integration of competitive BFN-based models like MolCraft with our feedback paradigm. Although multi-round feedback training improves performance, learning has not saturated; we will progressively increase the number of feedback rounds to probe the attainable upper bound. Moreover, the iterative-feedback paradigm is inherently more favorable to sampling-efficient models; for approaches with lower sampling efficiency (e.g., IPDiff), collecting preference pairs is substantially more time-consuming. Consequently, improving sampling efficiency for SBDD constitutes an important and promising avenue for future work.

# 7 ETHICS STATEMENT

This study does not involve any ethical issues, as all data used are either publicly available and anonymized or generated through model sampling, with no involvement of personal or sensitive information.

# 8 REPRODUCIBILITY STATEMENT

To ensure the reproducibility of our work, we provide a comprehensive description of the reward computation method, including normalization details in Appendix G. The process for constructing the preference pair dataset is thoroughly outlined in Section 5 and Algorithm 1. Furthermore, the model architectures and hyperparameter settings used in our experiments are detailed in Appendix H. We are committed to open-sourcing all code and trained models upon publication of this paper.

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

## A  THE USE OF LARGE LANGUAGE MODELS (LLMS)

This manuscript was prepared with the assistance of large language models (LLMs), which were used to improve the clarity and grammar of the content. The authors take full responsibility for all aspects of the manuscript, including any issues related to plagiarism and factual accuracy. No LLM is listed as an author of this work.

## B  DETAILED DERIVATION OF THE DIFFUSION-DPO OBJECTIVE

We give a concise derivation of the objective summarized in Sec. 4.2. Conditioning on the pocket $\mathcal{P}$ is omitted for brevity. The goal is to cast the per–timestep DPO preference comparison into a logistic objective whose margin combines continuous (coordinate) and discrete (atom/state) improvements of $p_\theta$ over a frozen reference $p_{\mathrm{ref}}$.

**Factorized reverse step.**  A single reverse step factorizes into continuous coordinates and discrete states:

$$p_\theta(\mathcal{L}_{t-1} \mid \mathcal{L}_t) = p_\theta^1(\mathbf{x}_{t-1} \mid \mathbf{x}_t)\, p_\theta^2(v_{t-1} \mid v_t), \tag{17}$$

with $p_{\mathrm{ref}}$ of the same form. For a preference pair $(\mathcal{L}_0^w \succ \mathcal{L}_0^l)$ the per–timestep Diffusion-DPO loss is

$$L_{\mathrm{DPO}}(\theta) = -\mathbb{E}_{t, \mathcal{L}_{t-1:t}^{w,l}} \left[ \log \sigma\Big( \beta T \big[ \log \tfrac{p_\theta(\mathcal{L}_{t-1}^w | \mathcal{L}_t^w)}{p_{\mathrm{ref}}(\mathcal{L}_{t-1}^w | \mathcal{L}_t^w)} - \log \tfrac{p_\theta(\mathcal{L}_{t-1}^l | \mathcal{L}_t^l)}{p_{\mathrm{ref}}(\mathcal{L}_{t-1}^l | \mathcal{L}_t^l)} \big] \Big) \right]. \tag{18}$$

This mirrors standard DPO: we apply a logistic loss to the difference of two log-likelihood improvements (winner vs. loser) relative to the reference.

**KL difference rewriting.**  Let $q(\mathcal{L}_{t-1} \mid \mathcal{L}_0, \mathcal{L}_t) = q^1(\mathbf{x}_{t-1} \mid \mathbf{x}_{0,t}) q^2(v_{t-1} \mid v_{0,t})$ be the forward posterior. Using the identity

$$\mathbb{E}_q \left[ \log \frac{p_\theta}{p_{\mathrm{ref}}} \right] = -\mathrm{KL}(q \| p_\theta) + \mathrm{KL}(q \| p_{\mathrm{ref}}),$$

each log-ratio is a (negative) KL improvement. Substituting for the winner and loser separately converts the margin inside $\log \sigma$ into a difference of improvements:

$$\log \frac{p_\theta(\mathcal{L}_{t-1}^w | \mathcal{L}_t^w)}{p_{\mathrm{ref}}(\cdot)} - \log \frac{p_\theta(\mathcal{L}_{t-1}^l | \mathcal{L}_t^l)}{p_{\mathrm{ref}}(\cdot)} = -\big[ \Delta_{\mathrm{KL}}^w - \Delta_{\mathrm{KL}}^l \big], \tag{19}$$

with

$$\Delta_{\mathrm{KL}}^* = \big( \mathrm{KL}(q^{1,*} \| p_\theta^1) - \mathrm{KL}(q^{1,*} \| p_{\mathrm{ref}}^1) \big) + \big( \mathrm{KL}(q^{2,*} \| p_\theta^2) - \mathrm{KL}(q^{2,*} \| p_{\mathrm{ref}}^2) \big), \quad * \in \{w, l\}. \tag{20}$$

Hence

$$L_{\mathrm{DPO}}(\theta) = -\mathbb{E} \left[ \log \sigma(-\beta T \cdot \Delta) \right], \qquad \Delta = \Delta_{\mathrm{KL}}^w - \Delta_{\mathrm{KL}}^l. \tag{21}$$

Intuitively, a positive $\Delta$ means the model reduces KL more (vs. reference) on the winner than on the loser, thus the logistic pushes probability mass toward such states.

**Continuous term.**  For fixed variance DDPM parameterized by noise prediction $\epsilon_\theta$,

$$\mathrm{KL}(q^1 \| p_\theta^1) - \mathrm{KL}(q^1 \| p_{\mathrm{ref}}^1) = \lambda_1(t) \Big( \| \epsilon - \epsilon_\theta(\mathcal{L}_t, t) \|_2^2 - \| \epsilon - \epsilon_{\mathrm{ref}}(\mathcal{L}_t, t) \|_2^2 \Big), \tag{22}$$

where schedule-dependent constants are absorbed into $\lambda_1(t)$. Thus improvement is exactly a reduction in noise-prediction squared error relative to the reference:

$$\Delta_\epsilon^* := \| \epsilon^* - \epsilon_\theta(\mathcal{L}_t^*, t) \|_2^2 - \| \epsilon^* - \epsilon_{\mathrm{ref}}(\mathcal{L}_t^*, t) \|_2^2.$$

A negative $\Delta_\epsilon^*$ indicates $p_\theta$ has better (lower) MSE than $p_{\mathrm{ref}}$ on that sample.

**Discrete term.**  Let $\pi(\mathbf{c}_t, \hat{\mathbf{c}}_0)$ be an exact (or proxy) posterior target for $v_{t-1}$. The categorical improvement is the difference of two cross-entropies (equivalently KL differences up to a shared entropy term):

$$\Delta_\pi^* := \sum_k \left[ p_\theta^2(k \mid \mathcal{L}_t^*) \log \frac{p_\theta^2(k \mid \mathcal{L}_t^*)}{\pi(\mathbf{c}_t^*, \hat{\mathbf{c}}_0^*)_k} - p_{\mathrm{ref}}^2(k \mid \mathcal{L}_t^*) \log \frac{p_{\mathrm{ref}}^2(k \mid \mathcal{L}_t^*)}{\pi(\mathbf{c}_t^*, \hat{\mathbf{c}}_0^*)_k} \right]. \tag{23}$$

If $p_\theta^2$ aligns better with $\pi$ than $p_{\mathrm{ref}}^2$, this term becomes negative (an improvement).

**Combined form.**

$$\Delta = \lambda_1(t)\big(\Delta_\epsilon^w - \Delta_\epsilon^l\big) + \lambda_2(t)\big(\Delta_\pi^w - \Delta_\pi^l\big), \qquad L_{\mathrm{DPO}}(\theta) = -\mathbb{E}\big[\log \sigma(-\beta T \Delta)\big]. \qquad (24)$$

Thus $\Delta$ is a weighted net advantage of $p_\theta$ over $p_{\mathrm{ref}}$ on the winner relative to the loser, combining continuous and discrete evidence. Note $\lambda_1$ and $\lambda_2$ are commonly set as constants in practice Ho et al. (2020); Kingma et al. (2021).

## C  RFFT PSEUDO-CODE

The pseudocode for dataset construction in our method is provided in Algorithm 1, while the training procedure is detailed in Algorithm 2. It is important to note that RFFT operates as an iterative framework, allowing Algorithm 1 and Algorithm 2 to be executed in multiple rounds of training.

---

**Algorithm 1** RFFT Dataset Construction pseudo-code

---

**Require:** Number of diffusion steps $T$, original training SBDD dataset $\mathcal{D}_{SBDD}$ with size $|\mathcal{D}|$, blank Winner-Loser dataset $\mathcal{D}_{wl}$, learning rate $\eta$, initial model parameters $\theta_0$.
1: Initialize model parameters $\theta \leftarrow \theta_0$
2: **for** $d = 1$ **to** $|\mathcal{D}|$ **do**
3:    **for** $n = 1$ **to** $N$ **do**
4:       Sample a data $(\mathcal{L}, \mathcal{P})$ from the $\mathcal{D}_{SBDD}$, record the Ligand $\mathcal{L}$ size as $|\mathcal{L}|$.
5:       Sample 2 initial states $\mathcal{L}_T^1$ and $\mathcal{L}_T^2$ with fixed size $|\mathcal{L}|$
6:       **for** $t = T - 1$ **to** $0$ **do**
7:          Compute prediction $\mathcal{L}_{t-1}^1 = f_\theta(\mathcal{L}_t^1)$ and $\mathcal{L}_{t-1}^2 = f_\theta(\mathcal{L}_t^2)$
8:       **end for**
9:       Calculate the chemical properties of $\mathcal{L}_0^1$ and $\mathcal{L}_0^2$ and their affinities to $\mathcal{P}$
10:      Determine to get the Winner and Loser $\mathcal{L}_0^w$ and $\mathcal{L}_0^l$.
11:      Add $(\mathcal{L}_0^w, \mathcal{L}_0^l, \mathcal{P})$ to $\mathcal{D}_{wl}$
12:    **end for**
13: **end for**

---

**Algorithm 2** Training RFFT pseudo-code

---

**Require:** Number of epochs $N$, number of steps $T$, Winner-Loser dataset $\mathcal{D}_{wl}$, learning rate $\eta$, initial model parameters $\theta_0$
1: Initialize model parameters $\theta_{\mathrm{ref}} \leftarrow \theta_0$, $\theta \leftarrow \theta_0$ and set the gradient of $\theta_{\mathrm{ref}}$ to be fixed
2: **for** $n = 1$ **to** $N$ **do**
3:    Shuffle dataset $\mathcal{D}$
4:    **for** each batch $B$ in $\mathcal{D}$ **do**
5:       **for** each data sample $(\mathcal{L}_0^w, \mathcal{L}_0^l, \mathcal{P})$ in $\mathcal{D}_{wl}$ **do**
6:          Sample time step $t$ from $[1, T-1]$
7:          Sample $\mathcal{L}_t^w$ and $\mathcal{L}_t^l$ using the same noise $\epsilon$
8:          Compute the $L(\theta) = -\mathbb{E}_{((\mathbf{x}_0^w, v_0^w),(\mathbf{x}_0^l, v_0^l)),t,\mathbf{x}_{t-1,t}^{w,l}, v_{t-1,t}^{w,l}} \left[\log \sigma(-\beta T \cdot \Delta)\right]$
9:          Update parameters: $\theta \leftarrow \theta - \eta \nabla_\theta \mathcal{L}$
10:      **end for**
11:    **end for**
12: **end for**

---

## D  DIFFUSION MODELS FOR CONTINUOUS DATA

Diffusion models operate by gradually introducing noise to samples drawn from the data distribution $q(\mathbf{x}_0)$ according to a predefined schedule. This forward process progressively transforms the data distribution into a prior distribution $q(\mathbf{x}_T)$. During inference, the generative process employs a denoising diffusion approach via reverse Markov sampling as formulated in Ho et al. (2020); Song et al. (2021):

$$p_\theta(\mathbf{x}_{t-1}|\mathbf{x}_t) = \mathcal{N}\left(\mathbf{x}_{t-1}; \boldsymbol{\mu}_\theta(\mathbf{x}_t, t), \frac{\sigma_{t|t-1}^2 \sigma_{t-1}^2}{\sigma_t^2}\mathbf{I}\right). \tag{25}$$

The training objective is formulated as minimizing the Evidence Lower Bound (ELBO), which simplifies to minimizing the KL divergence at each timestep:

$$\sum_{t=2}^{T} D_{\mathrm{KL}}\left(q(\mathbf{x}_{t-1}|\mathbf{x}_t, \mathbf{x}_0)|p_\theta(\mathbf{x}_{t-1}|\mathbf{x}_t)\right). \tag{26}$$

Specifically, the single-step loss term $L_t$ can equivalently be expressed in two parameterizations:

$$L_t = \mathbb{E}_{\mathbf{x}_0, \boldsymbol{\epsilon}, t}\left[\omega_1(\lambda_t)|\boldsymbol{\mu}_t(\mathbf{x}_t, \mathbf{x}_0) - \boldsymbol{\mu}_\theta(\mathbf{x}_t, t)|^2\right] = \mathbb{E}_{\mathbf{x}_0, \boldsymbol{\epsilon}, t}\left[\omega_2(\lambda_t)|\boldsymbol{\epsilon}_t - \boldsymbol{\epsilon}_\theta(\mathbf{x}_t, t)|^2\right], \tag{27}$$

where $\boldsymbol{\epsilon} \sim \mathcal{N}(0, \mathbf{I})$ denotes standard Gaussian noise, $t \sim \mathcal{U}(0, T)$ represents uniformly sampled timesteps, and the noisy sample at timestep $t$ is drawn from:

$$q(\mathbf{x}_t|\mathbf{x}_0) = \mathcal{N}\left(\mathbf{x}_t; \alpha_t \mathbf{x}_0, \sigma_t^2 \mathbf{I}\right). \tag{28}$$

Here, $\alpha_t$ and $\sigma_t$ define the noise schedule as described in Rombach et al. (2022). The parameter $\lambda_t = \alpha_t^2/\sigma_t^2$ represents the signal-to-noise ratio (SNR) Kingma et al. (2021), and the weighting functions $\omega_1(\lambda_t)$ and $\omega_2(\lambda_t)$ are typically chosen as constants following Ho et al. (2020).

# E  DIFFUSION MODELS FOR DISCRETE DATA

Multinomial diffusion models extend continuous diffusion frameworks to categorical data, where each discrete data point $\mathbf{c}_t \in \{0, 1\}^K$ is represented as a one-hot vector Hoogeboom et al. (2021); Guan et al. (2023a). The forward process gradually introduces uniform noise across the $K$ categories:

$$q(\mathbf{c}_t|\mathbf{c}_{t-1}) = \mathcal{C}\left(\mathbf{c}_t; (1 - \beta_t)\mathbf{c}_{t-1} + \frac{\beta_t}{K}\mathbf{1}\right), \tag{29}$$

where $\mathcal{C}$ denotes a categorical distribution. The marginal distribution at timestep $t$ given $\mathbf{c}_0$ is:

$$q(\mathbf{c}_t|\mathbf{c}_0) = \mathcal{C}\left(\mathbf{c}_t; \bar{\alpha}_t \mathbf{c}_0 + \frac{1 - \bar{\alpha}_t}{K}\mathbf{1}\right), \tag{30}$$

with $\alpha_t = 1 - \beta_t$ and $\bar{\alpha}_t = \prod_{\tau=1}^{t} \alpha_\tau$. The posterior distribution can be computed in closed form:

$$q(\mathbf{c}_{t-1}|\mathbf{c}_t, \mathbf{c}_0) = \mathcal{C}\left(\mathbf{c}_{t-1}; \boldsymbol{\theta}_{\mathrm{post}}(\mathbf{c}_t, \mathbf{c}_0)\right), \tag{31}$$

where

$$\boldsymbol{\theta}_{\mathrm{post}}(\mathbf{c}_t, \mathbf{c}_0) = \frac{\tilde{\boldsymbol{\theta}}}{\sum_{k=1}^{K} \tilde{\theta}_k}, \quad \tilde{\boldsymbol{\theta}} = \left[\alpha_t \mathbf{c}_t + \frac{1 - \alpha_t}{K}\mathbf{1}\right] \odot \left[\bar{\alpha}_{t-1}\mathbf{c}_0 + \frac{1 - \bar{\alpha}_{t-1}}{K}\mathbf{1}\right]. \tag{32}$$

For the generative process, we predict the original data point as $\hat{\mathbf{c}}_0 = \mu\theta(\mathbf{c}_t, t)$, and define the reverse transition probability as:

$$p_\theta(\mathbf{c}_{t-1}|\mathbf{c}_t) = \mathcal{C}\left(\mathbf{c}_{t-1}; \boldsymbol{\theta}_{\mathrm{post}}(\mathbf{c}_t, \hat{\mathbf{c}}_0)\right). \tag{33}$$

Following the continuous case shown in Eq. (26), the training objective for discrete diffusion models is formulated by minimizing the KL divergence between the true posterior and the model's approximation at each timestep:

$$D_{\mathrm{KL}}\left(q(\mathbf{c}_{t-1}|\mathbf{c}_t, \mathbf{c}_0)|p_\theta(\mathbf{c}_{t-1}|\mathbf{c}_t)\right) = D_{\mathrm{KL}}\left(\mathcal{C}(\boldsymbol{\theta}_{\mathrm{post}}(\mathbf{c}_t, \mathbf{c}_0))|\mathcal{C}(\boldsymbol{\theta}_{\mathrm{post}}(\mathbf{c}_t, \hat{\mathbf{c}}_0))\right). \tag{34}$$

Explicitly, this KL divergence is computed as:

$$D_{\mathrm{KL}} = \sum_{k=1}^{K} \boldsymbol{\theta}_{\mathrm{post}}(\mathbf{c}_t, \mathbf{c}_0)_k \cdot \log \frac{\boldsymbol{\theta}_{\mathrm{post}}(\mathbf{c}_t, \mathbf{c}_0)_k}{\boldsymbol{\theta}_{\mathrm{post}}(\mathbf{c}_t, \hat{\mathbf{c}}_0)_k}. \tag{35}$$

where $\mathbf{c}_0$ and $\hat{\mathbf{c}}_0$ represent the real initial state and the predicted initial state.

# F  LIGAND SIZE AND GRADIENT VARIANCE IN DIFFUSION-BASED DPO TRAINING

Controlling ligand size is important in diffusion-based Direct Preference Optimization (DPO) training, as it directly impacts the variance of the gradients used for optimization. Here, we provide a detailed mathematical explanation for why controlling ligand size reduces gradient variance.

## F.1  DPO LOSS AND ITS GRADIENT

Follow Eq. 18, the DPO loss is defined as:

$$L_{\text{DPO}}(\theta) = -\mathbb{E}\left[\log \sigma(\beta_T \cdot \Delta)\right] \tag{36}$$

where $\sigma(\cdot)$ is the sigmoid function, $\beta_T$ is a temperature parameter, and $\Delta$ is given by:

$$\Delta = \log \frac{p_\theta(L_{t-1}^w | L_t^w, P)}{p_{\text{ref}}(L_{t-1}^w | L_t^w, P)} - \log \frac{p_\theta(L_{t-1}^l | L_t^l, P)}{p_{\text{ref}}(L_{t-1}^l | L_t^l, P)} \tag{37}$$

where $L^w$ and $L^l$ denote the winner and loser ligands, respectively.

The gradient of the loss with respect to $\theta$ is:

$$\nabla_\theta L_{\text{DPO}} = -\mathbb{E}\left[\sigma'(\beta_T \Delta) \cdot \beta_T \cdot \nabla_\theta \Delta\right] \tag{38}$$

where $\sigma'(\cdot)$ is the derivative of the sigmoid function.

## F.2  DECOMPOSITION BY MOLECULAR SIZE

For molecular graphs, the log-probability can be decomposed atom-wise:

$$\log p_\theta(L_{t-1} | L_t, P) = \sum_{i=1}^{|L|} \log p_\theta(x_i, v_i | L_t, P) \tag{39}$$

where $|L|$ is the number of atoms in ligand $L$, and $(x_i, v_i)$ are the features of atom $i$.

Thus, the gradient of $\Delta$ becomes:

$$\nabla_\theta \Delta = \sum_{i=1}^{|L^w|} g_i^w - \sum_{j=1}^{|L^l|} g_j^l \tag{40}$$

where $g_i^w$ and $g_j^l$ are the per-atom gradients for the winner and loser molecules, respectively.

## F.3  GRADIENT VARIANCE CALCULATION

Assume the per-atom gradients $g$ have mean $\mu_g$ and variance $\sigma_g^2$. Then, the variance of $\nabla_\theta \Delta$ is:

$$\text{Var}\left[\nabla_\theta \Delta\right] = \text{Var}\left[\sum_{i=1}^{|L^w|} g_i^w - \sum_{j=1}^{|L^l|} g_j^l\right] \tag{41}$$

$$= |L^w|\sigma_g^2 + |L^l|\sigma_g^2 + \text{Var}\left[(|L^w| - |L^l|)\mu_g\right] \tag{42}$$

$$= (|L^w| + |L^l|)\sigma_g^2 + (|L^w| - |L^l|)^2 \mu_g^2 \tag{43}$$

The key term here is the quadratic dependence on the size difference:

$$(|L^w| - |L^l|)^2 \mu_g^2 \tag{44}$$

This term grows rapidly as the difference in ligand sizes increases.

### F.4 CONCLUSION

When ligand sizes are not controlled (i.e., $|L^w| \neq |L^l|$), the variance of the gradient $\nabla_\theta \Delta$ increases quadratically with the size difference. This increased variance can destabilize training and slow down convergence. Therefore, controlling ligand size—by matching sizes or restricting their range—directly reduces the gradient variance, leading to more stable and efficient optimization in diffusion-based DPO training.

## G NORMALIZATION OF THE CHEMICAL SCORES AND AFFINITY SCORE

To enable fair comparison and combination of different chemical property scores and affinity scores, we analyzed the distribution of these scores in the original CrossDock2020 dataset Francoeur et al. (2020) and applied min-max normalization. Specifically, the affinity scores were computed using AutoDock Vina in "Score Only" mode Trott & Olson (2010). Vina scores greater than 0, which indicate no binding interaction, were set to 0. The detailed normalization ranges are summarized in Table 4.

Table 4: Normalization Ranges for Reward Components

| Property | Min Value | Max Value |
|---|---|---|
| QED | 0.0154 | 0.9475 |
| LogP | -15.2306 | 15.6260 |
| SA | 0.1800 | 1.0000 |
| Lipinski | 0 | 5 |
| Affinity | -17.3050 | 0 |

## H DETAILED MODEL TRAINING SETTINGS AND COMPUTATIONAL RESOURCES

We adopt the TargetDiff architecture Guan et al. (2023a) and the Diffusion-DPO algorithm in our experiments. The pretrained TargetDiff model is provided by Lin et al. (2025b). The model comprises nine EGNN Satorras et al. (2021) layers with GAT Veličković et al. (2018) and a hidden dimension of 128. We set the number of diffusion steps to 1000. For the alignment training stage, we use a learning rate of $1 \times 10^{-6}$, clip the gradient norm at 0.0001, set the batch size to 2, and train for 5 epochs. The hyperparameters are: $\beta \in \{2, 5\}$ ($\beta = 5$ in main text), $\lambda_1(t) = 1$, and $\lambda_2(t) \in \{1, 100\}$ ($\lambda_2(t) = 1$ in main text). We use the Adam optimizer Kingma & Ba (2014) for training. All experiments are conducted on a Tesla V100S-PCIe-32GB GPU.

## I DETAILED METRIC DEFINITIONS

We define the metrics used in the main text, following Lin et al. (2025b).

### I.1 JENSEN-SHANNON DIVERGENCE (JSD)

The Jensen-Shannon divergence (JSD) between two probability distributions $P$ and $Q$ is defined as:

$$M = \frac{1}{2}(P + Q)$$

$$KL(P\|M) = \sum_{x \in \mathcal{X}} P(x) \log \frac{P(x)}{M(x)}$$

$$KL(Q\|M) = \sum_{x \in \mathcal{X}} Q(x) \log \frac{Q(x)}{M(x)} \tag{45}$$

$$JSD(P\|Q) = \frac{1}{2} KL(P\|M) + \frac{1}{2} KL(Q\|M)$$

where $P$ is the reference distribution, $Q$ is the model-generated distribution, and $M$ is the mixture distribution.

### I.2 MEAN ABSOLUTE ERROR (MAE)

For a set of substructure types $S$, the mean absolute error (MAE) is defined as:

$$\text{MAE} = \frac{1}{|S|} \sum_{i \in S} \left| f_i^{\text{gen}} - f_i^{\text{ref}} \right| \tag{46}$$

where $f_i^{\text{gen}}$ and $f_i^{\text{ref}}$ denote the frequencies of substructure $i$ in the generated and reference molecules, respectively.

### I.3 IMPROVEMENT PERCENTAGE (IMP%)

Improvement Percentage (IMP%) measures the fraction of generated molecules whose binding affinity (e.g., docking score) is better (lower) than that of the reference molecule.

$$\text{IMP\%} = \frac{N_{\text{improved}}}{N_{\text{total}}} \times 100\% \tag{47}$$

where $N_{\text{improved}}$ is the number of generated molecules with better (lower) binding affinity than the reference molecule, and $N_{\text{total}}$ is the total number of generated molecules.

### I.4 MEAN PERCENTAGE BETTER GAP (MPBG%)

MPBG quantifies the average percentage improvement in binding energy relative to the reference molecule.

$$\text{MPBG\%} = \frac{1}{N} \sum_{i=1}^{N} \left( \frac{E_{i,\text{gen}} - E_{\text{ref}}}{E_{\text{ref}}} \times 100\% \right) \tag{48}$$

where $E_{i,\text{gen}}$ is the binding energy of the $i$-th generated molecule, $E_{\text{ref}}$ is the binding energy of the reference molecule, and $N$ is the number of generated molecules.

### I.5 LIGAND BINDING EFFICIENCY (LBE)

LBE measures the average binding energy contribution per ligand atom.

$$\text{LBE}_i = -\frac{E_{i,\text{gen}}}{N_{i,\text{lig}}} \tag{49}$$

where $E_{i,\text{gen}}$ is the binding energy of the $i$-th generated molecule, and $N_{i,\text{lig}}$ is the number of atoms in that molecule.

## J    RATIONALE FOR USING WEIGHTED REWARD SCORE AS REWARD FUNCTION

In SBDD, the goal is to generate molecules that simultaneously optimize multiple chemical properties and binding affinities. Given the complexity of balancing these competing objectives, we employed a weighted reward function that combines various molecular properties with binding affinity scores.

While certain chemical properties exhibit correlations—such as the positive relationship between synthetic accessibility (SA) and quantitative estimate of drug-likeness (QED) reported in Cremer et al. (2024)—this does not guarantee simultaneous optimization of all desirable traits. Previous studies have demonstrated inherent trade-offs in multi-objective molecular optimization. For instance, MARS Xie et al. (2021) showed that separate optimization of QED and SA leads to significant degradation in the other metric when improving one property. Similarly, Pareto frontier analysis in Hömberg et al. (2024) revealed that enhancing QED typically results in decreased SA values, confirming the existence of fundamental trade-offs between these properties.

Given that SBDD requires concurrent consideration of multiple molecular attributes rather than isolated optimization, our weighted reward approach provides a practical solution for navigating these trade-offs.

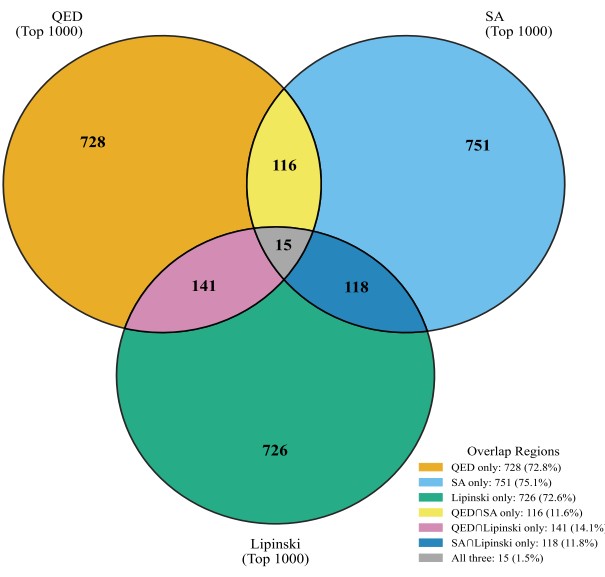

Figure 4: Empirical Analysis of Molecular Property Correlations: Overlap Distribution of Top-Performing Ligands (Analysis of 11,740 Total Ligands)

To empirically validate these trade-offs, we analyzed the top 1000 ligands from our training set of 11,740 molecules across QED, SA, and Lipinski scores, examining their pairwise and collective overlaps as illustrated in Figure 4. The results reveal limited overlap between high-performing sets: only 131 molecules appear in both top QED and SA categories, 156 in QED and Lipinski, and 133 in SA and Lipinski. Strikingly, only 15 ligands achieve top performance across all three metrics simultaneously. This analysis provides concrete evidence that despite observed correlations, these properties cannot be universally optimized together.

To further quantify these trade-offs, we compared the top 1000 QED performers (set S1) with the top 1000 SA performers (set S2). The QED distributions show significant divergence: S1 achieves a mean QED of 0.8698 ± 0.0323, while S2 has a substantially lower mean of 0.6607 ± 0.1436. Conversely, SA scores reveal the opposite pattern: S1 has a mean SA of 0.7897 ± 0.0854, compared to S2's superior mean of 0.9111 ± 0.0230. These statistical differences provide definitive

evidence that QED and SA optimization represents fundamentally competing objectives that cannot be simultaneously maximized across all molecules.

## K  MODEL GENERALIZATION STUDY

To validate the generalization capability of our proposed RFFT approach, we conduct comprehensive experiments across different model architectures, hyperparameter configurations, and reward function settings. The experimental results demonstrate consistent performance improvements in three key dimensions: (1) different backbone architectures, (2) different hyperparameter settings, and (3) different reward function formulations.

### K.1  GENERALIZATION ACROSS DIFFERENT BACKBONE ARCHITECTURES

We evaluate RFFT on two distinct diffusion models: TargetDiff and IPDiff Huang et al. (2024). The successful application of RFFT to both models provides validation of our method's generalization capability and broad applicability to various diffusion-based molecular generation frameworks.

### K.2  GENERALIZATION ACROSS DIFFERENT HYPERPARAMETER SETTINGS

We investigate the robustness of our method under different hyperparameter configurations, specifically varying the temperature parameter $\beta$ in $\{2, 5\}$. The results show consistent improvements across different $\beta$ values, demonstrating the stability of our approach.

### K.3  GENERALIZATION ACROSS DIFFERENT REWARD FUNCTION CONFIGURATIONS

We examine the performance of RFFT under different reward function formulations. In addition to the main reward function used in the paper:

$$R = 0.125 \times (\text{QED} + \text{SA} + \text{LogP} + \text{Lipinski}) + 0.5 \times (-\text{Vina score}) \qquad (50)$$

We also incorporate Boltz-2 Passaro et al. (2025) predictions as an additional reward component. Boltz-2 predicts protein-ligand interactions similar to re-docking scenarios, which aligns well with the improvements observed in both Vina-min and Vina-dock modes. The enhanced reward function is formulated as:

$$\text{reward} = 0.125 \times (\text{QED} + \text{SA} + \text{LogP} + \text{Lipinski}) + 0.3 \times (-\text{Vina score}) + 0.2 \times (-\log(\text{IC}_{50})) \quad (51)$$

For the Boltz-2 enhanced experiments, we used 360 labeled winner/loser pairs to train the model.

The experimental results are presented in Table 5, demonstrating the consistent performance improvements achieved by RFFT across all three dimensions.

Table 5: Model Generalization Study Results

| Method | QED | SA | LPSK | Validity% | Vina Score | | Vina Min | | Vina Dock | | | |
| | | | | | $E_{\text{vina}}$ | IMP% | $E_{\text{vina}}$ | IMP% | $E_{\text{vina}}$ | IMP% | MPBG% | LBE |
| *Different Backbone Architectures* | | | | | | | | | | | | |
| TargetDiff | 0.487 | 0.598 | 4.57 | 97.05 | -5.73 | 38.51 | -6.44 | 47.14 | -7.39 | 52.17 | 5.53 | 0.3545 |
| TargetDiff-RFFT($\beta = 5$) | 0.49 | 0.607 | 4.59 | 96.47 | -6.1 | 43.5 | -6.68 | 50.78 | -7.6 | 55.37 | 8.26 | 0.3614 |
| IPDiff | 0.465 | 0.576 | 4.348 | 93.22 | -4.32 | 36.85 | -5.79 | 45.84 | -7.38 | 51.86 | 5.1 | 0.3592 |
| IPDiff-RFFT($\beta = 5$) | 0.471 | 0.586 | 4.388 | 92.46 | -5.66 | 45.41 | -6.42 | 51.5 | -7.61 | 55.19 | 7.21 | 0.365 |
| *Different Hyperparameter Settings* | | | | | | | | | | | | |
| TargetDiff-RFFT($\beta = 2$) | 0.493 | 0.599 | 4.59 | 97.4 | -6.08 | 44 | -6.67 | 51.07 | -7.54 | 54.73 | 7.11 | 0.357 |
| TargetDiff-RFFT($\beta = 5$) | 0.49 | 0.607 | 4.59 | 96.47 | -6.1 | 43.5 | -6.68 | 50.78 | -7.6 | 55.37 | 8.26 | 0.3614 |
| *Different Reward Functions* | | | | | | | | | | | | |
| TargetDiff-RFFT($\beta = 5$, Standard) | 0.49 | 0.607 | 4.59 | 96.47 | -6.1 | 43.5 | -6.68 | 50.78 | -7.6 | 55.37 | 8.26 | 0.3614 |
| TargetDiff-RFFT($\beta = 5$, with Boltz-2) | 0.487 | 0.599 | 4.57 | 97.33 | -5.96 | 40.16 | -6.67 | 50.16 | -7.54 | 54.97 | 7.46 | 0.3581 |

The results clearly demonstrate that RFFT achieves consistent improvements across all experimental dimensions. Compared with the baseline results, we observe significant enhancements in binding affinity metrics (Vina scores, IMP%, MPBG%) while maintaining or improving molecular quality metrics (QED, SA, Lipinski). The successful generalization to different backbone architectures

1134
1135
1136
1137
1138
1139
1140
1141
1142
1143
1144
1145
1146
1147
1148
1149
1150
1151
1152
1153
1154
1155
1156
1157
1158
1159
1160
1161
1162
1163
1164
1165
1166
1167
1168
1169
1170
1171
1172
1173
1174
1175
1176
1177
1178
1179
1180
1181
1182
1183
1184
1185
1186
1187

(TargetDiff and IPDiff) validates the broad applicability of our method, while the consistent performance across different hyperparameter settings and reward function configurations demonstrates its robustness and flexibility.

## L    ADDITIONAL CASE STUDY EXAMPLES

To further demonstrate the effectiveness of RFFT, we provide additional case study examples that complement the main text analysis. Figure 5 shows four randomly selected ligand pairs from 3 test pocket which are "5w2g", "2v3r", and "5bur", where both TargetDiff and TargetDiff-RFFT are sampled with identical noise seeds and inference settings. These examples illustrate the consistent improvement in molecular quality and binding affinity achieved through our reward-based fine-tuning approach.

The examples in Figure 5 further validate our findings from the main ablation study, showing that RFFT consistently generates molecules with improved chemical properties and better binding affinity compared to the baseline TargetDiff model across different molecular structures and binding contexts.

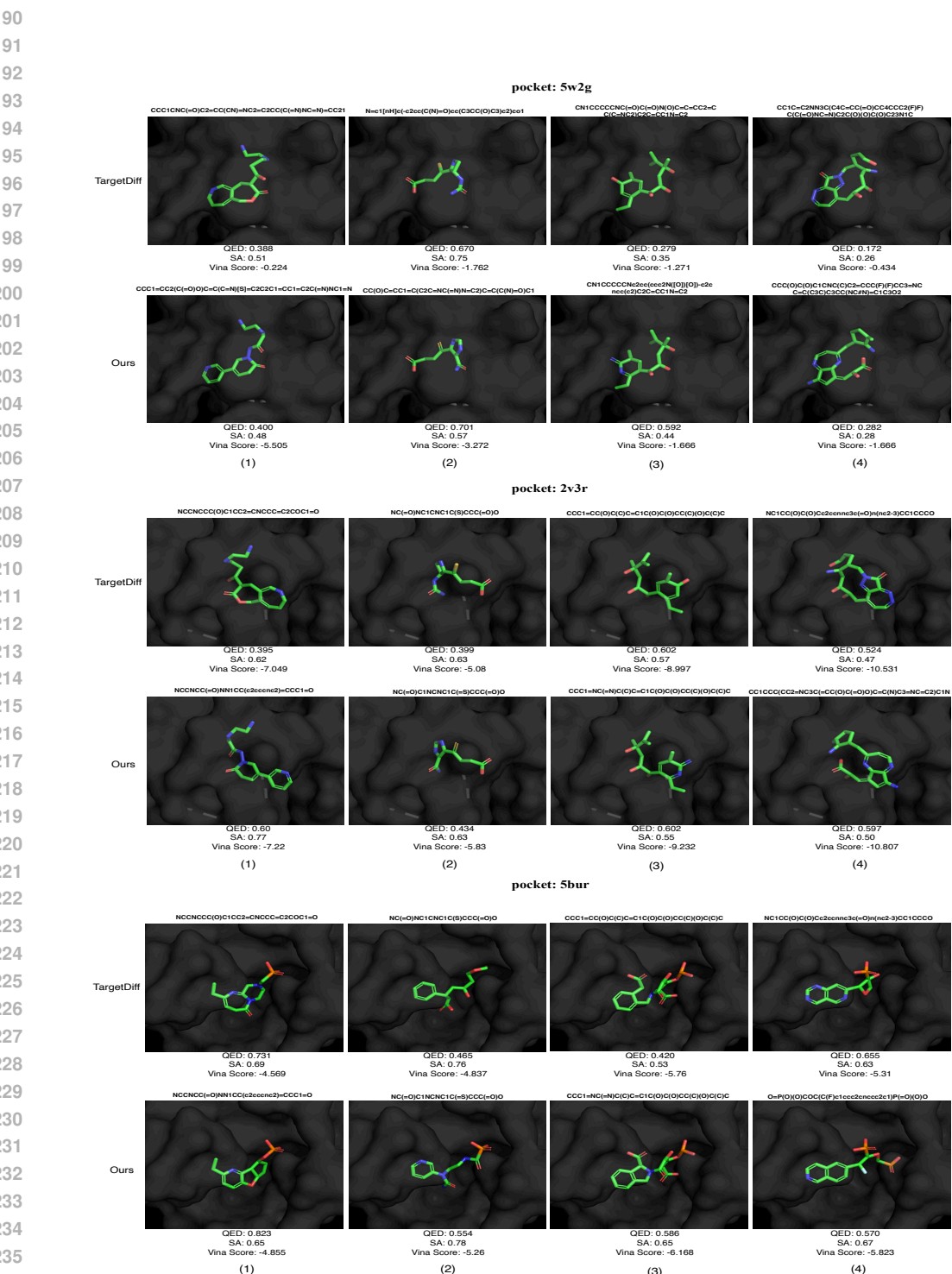

Figure 5: Additional case study examples.

