# OpenReview forum: "Reward-Focused Fine-tuning of Pocket-aware Diffusion Models via Direct Preference Optimization"
_ICLR.cc/2026/Conference — ICLR 2026 Conference Withdrawn Submission_

### Official Review · Reviewer_uonK · 2025-10-30

**Soundness:** 3
**Presentation:** 2
**Contribution:** 2
**Rating:** 6
**Confidence:** 3

**Summary:**

This paper introduces ​​Reward-Focused Fine-Tuning (RFFT)​​, a framework for optimizing pocket-specific molecule generation, which applies ​​Direct Preference Optimization (DPO)​​ to diffusion models like TargetDiff, creating a feedback loop for iterative model enhancement.

The method uses the model's ​​own generated samples​​ and a reward function for fine-tuning, rather than relying solely on the original training dataset. Crucially, the authors demonstrate that this approach is highly data-efficient; fine-tuning with a small number of self-generated samples can surprisingly outperform using a larger sample set from the original data.

Experimental results confirm that ​​RFFT achieves superior performance​​ across critical metrics, especially the binding affinity.

**Strengths:**

- **Clear and Well-Structured Presentation**: Figure 1 effectively illustrates the pipeline of the proposed framework, making the methodology easy to follow.
- **Simple yet Effective Approach**: The experiments demonstrate that the method successfully iteratively improves the properties of generated molecules, particularly in binding affinity (as shown in Section 5.2).  It is also shown that using self-generated molecules as preference pairs is more effective than using the molecules in the original dataset, as done in AliDiff.
- **Unexpected Benefit in Substructure Fidelity**: Interestingly, the fine-tuned model generates molecules with substructure distributions slightly closer to the reference, despite this not being explicitly encoded in the reward function.
- **Flexibility in Reward Design**: The use of *Boltz2*-predicted protein-ligand interaction as a reward showcases the method’s adaptability to diverse reward formulations, highlighting its generalization potentia (as shown in the Appendix)l.
- **Sufficiently Discussed Limiations**: The limitations are sufficiently discussed in Section 6.


References:

[1] Stabilizing Policy Gradients for Stochastic Differential Equations via Consistency with Perturbation Process, ICML 2024

[2] Antigen-Specific Antibody Design via Direct Energy-based Preference Optimization, NeurIPS 2024

[3] Decomposed Direct Preference Optimization for Structure-Based Drug Design, TMLR 2025

[4] Enhancing Ligand Validity and Affinity in Structure-Based Drug Design with Multi-Reward Optimization, ICML 2025

**Weaknesses:**

1. **Lack of Quantitative Diversity Analysis**
   - While Section 5.3 qualitatively observes differences in molecular diversity across pockets post fine-tuning, the work lacks a **quantitative assessment** of diversity (e.g., using metrics like Tanimoto similarity or scaffold diversity).
   - A critical analysis of the **reward-diversity trade-off** is missing, which is essential for evaluating the practical utility of the method.

2. **Incomplete Coverage of Related Work**
   The submission overlooks several closely related studies that similarly employ DPO or RL for molecular optimization:
   - **[1]**: Uses RL to fine-tune TargetDiff for ligand property optimization in structure-based drug design (SBDD).
   - **AbDPO [2]**: Applies DPO to diffusion models for biomolecular design (antibodies), differing only in target (proteins vs. small molecules).
   - **DecompDPO [3]**: Directly uses preference optimization on self-generated ligands, yet is inadequately cited.
   - **[4]**: Optimizes Bayesian Flow Networks (BFNs) with DPO for SBDD, a closely related approach.
   These omissions weaken the contextualization of the work’s contributions.

3. **Marginal Novelty**
   - The core idea, **DPO on self-generated samples for SBDD**, was already introduced in *DecompDPO [3]*, making the methodological novelty incremental.
   - While the focus on small-molecule ligands differs from prior work (e.g., AbDPO’s antibodies), the fundamental framework remains largely unchanged.

**Questions:**

The questions have been listed in the above weaknesses. My current score is marginally positive, considering the interesting experimental results. However, this assessment is provisional. As outlined in the Weaknesses section, significant concerns regarding novelty and the coverage of related work remain. My final score is contingent on the authors' rebuttal and is subject to change based on their ability to adequately address these points, particularly by providing a thorough discussion to distinguish their contribution from the cited related works.

**Details Of Ethics Concerns:**

This paper studies fine-tuning generative models for structure-based drug design and only focuses on in silico experiments. Thus there is **no ethics concerns**.

---

### Official Review · Reviewer_nx8b · 2025-11-01

**Soundness:** 2
**Presentation:** 2
**Contribution:** 2
**Rating:** 2
**Confidence:** 4

**Summary:**

This paper tackles the task of structure-based drug design (SBDD) and proposes **Reward-Focused Fine-Tuning (RFFT)** for fine-tuning protein pocket-conditioned small molecule generative models, using in-silico molecule quality rewards. In particular, the paper suggests to take a given diffusion-based generative model -- in this case TargetDiff -- and sample ligand pairs of the same size for given pockets. Then, the two molecules are scored and a winner-loser preference pair is created. That way, a preference dataset is created. Next, the model is fine-tuned using the created preference data via diffusion direct preference optimization (DPO). The process is repeated over several rounds to further improve the model. The authors validate the proposed RFFT method by comparing their fine-tuned TargetDiff model to non-fine-tuned baselines and via ablation studies. They show favourable results, on-par with existing state-of-the-art models.

**Strengths:**

**Clarity:** The proposed method is simple and the paper is generally easy to understand and follow.

**Experiments:** The paper runs extensive experiments and compares the proposed approach to various baselines, on many different metrics, showing results either outperforming baselines or being on par with existing state-of-the-art methods.

**Originality and Significance:** The task tackled by this paper, structure-based drug design, is an impactful application and the method shows results on par with other state-of-the-art approaches. I have concerns regarding originality and significance, though, see below.

**Weaknesses:**

There are various weaknesses:

**Originality:** Overall, the paper's proposed approach is a simple, repeated application of Diffusion DPO to existing SBDD diffusion models with repeated self-generation of the preference dataset. There is no significant technical or methodological novelty in the paper.

**AliDiff relation and comparison:** As the authors discuss in their related work section, the method is most similar to AliDiff [1], which also used Diffusion DPO to fine-tune SBDD diffusion models -- hence, this alone is not new anymore. The authors then do correctly point out that AliDiff fine-tunes on existing data, and not on self-generated data, though. Nonetheless, it seems that AliDiff is publicly available, and given the close relation to RFFT, this seems like the most important baseline to compare to. Despite that, there is no comparison to AliDiff.

**Diversity analysis:** Reward-based fine-tuning can enhance quality, as observed in the paper, but it can also negatively impact the diversity of the data generated by the model. However, there is no quantitative evaluation of the diversity of the generated molecules. Previous work, such as the mentioned AliDiff [1], also reported diversity.

**Overhead due to preference data creation:** While fine-tuning on self-generated data does enhance performance, the repeated self-generation of the preference data implies substantial computational overhead. This is a weakness of the method, and the authors did not discuss this or specify how fast or slow this is in practice.

**Experimental results:** While the proposed RFFT method improves performance compared to the non-fine-tuned TargetDiff model, some improvements are incremental and the paper overall is still not significantly better than MolCraft.

**Conclusions:** The limited novelty and originality as well as the missing comparisons and analyses are significant concerns and the experimental results are okay, but do not represent to a major leap. Therefore, I do not think this paper meets the bar for acceptance.

[1] Gu et al., "Aligning Target-Aware Molecule Diffusion Models with Exact Energy Optimization", NeurIPS, 2024, https://github.com/MinkaiXu/AliDiff

**Questions:**

- When creating the preference data, why are only 80% of the winner-loser pairs used?
- What happens when both molecules that are generated as part of the preference data creation have low quality? The method will then create a winner-loser pair from two ligands that both have low quality, which seems not useful for fine-tuning.

---

### Official Review · Reviewer_7GhC · 2025-11-02

**Soundness:** 1
**Presentation:** 2
**Contribution:** 1
**Rating:** 2
**Confidence:** 4

**Summary:**

This paper proposes Reward-Focused Fine-Tuning, a framework that iteratively fine-tunes pocket-conditioned diffusion models for structure-based drug design using Direct Preference Optimization. The model generates ligand pairs, assigns reward-based winner–loser labels, and fine-tunes the diffusion model in multiple rounds to enhance ligand quality and binding affinity. The authors demonstrate experiments on TargetDiff, reporting gains in chemical property metrics and docking-based affinity measures compared to their baselines.

**Strengths:**

- The iterative DPO fine-tuning idea is conceptually reasonable and aligns with recent trends in preference-based model alignment.
- The writing is overall organized and easy to follow, and the paper provides clear descriptions of the iterative process and loss formulation.

**Weaknesses:**

- Lack of novelty. The proposed method is a direct application of existing preference-alignment techniques. Self-improvement with DPO is a straightforward idea already explored in works such as  Molform[1], DecompDPO[2], and AliDiff[3]. The technical novelty beyond applying DPO is minimal.
- Unfair comparison. The paper positions RFFT as an optimization method but only compares against generative baselines. Without including any recent optimization-based methods, the evaluation does not support the claim that RFFT achieves superior fine-tuning performance and raises concerns about the fairness and completeness of the experimental validation.
- Missing baselines. The comparison excludes several recent and relevant baselines, which at least should including generative methods such as DecompDiff[4] and MolCRAFT[5], optimization methods such as TAGMol[6], MolJO[7], AliDiff[3], and DecompDPO[2].
- Limited improvement and overstated claims. The reported performance gains are marginal, especially for chemical property metrics (QED, SA, Lipinski) where improvements appear within noise level. The claimed state-of-the-art performance is not convincing given the small numerical differences and absence of stronger recent baselines.

[1] Huang, J., & Zhang, D. MolFORM: Multi-modal Flow Matching for Structure-Based Drug Design. In ICML 2025 Generative AI and Biology (GenBio) Workshop.

[2] Cheng, X., Zhou, X., Yang, Y., Bao, Y., & Gu, Q. Decomposed Direct Preference Optimization for Structure-Based Drug Design. In  Transactions on Machine Learning Research.

[3] Gu, S., Xu, M., Powers, A. S., Nie, W., Geffner, T., Kreis, K., ... & Ermon, S. Aligning Target-Aware Molecule Diffusion Models with Exact Energy Optimization. In The Thirty-eighth Annual Conference on Neural Information Processing Systems.

[4] Guan, J., Zhou, X., Yang, Y., Bao, Y., Peng, J., Ma, J., ... & Gu, Q. (2023, July). DecompDiff: Diffusion Models with Decomposed Priors for Structure-Based Drug Design. In International Conference on Machine Learning (pp. 11827-11846). PMLR.

[5] Qu, Y., Qiu, K., Song, Y., Gong, J., Han, J., Zheng, M., ... & Ma, W. Y. MolCRAFT: Structure-Based Drug Design in Continuous Parameter Space. In Forty-first International Conference on Machine Learning.

[6] Dorna, V., Subhalingam, D., Kolluru, K., Tuli, S., Singh, M., Singal, S., ... & Ranu, S. TAGMol: Target-Aware Gradient-guided Molecule Generation. In ICML'24 Workshop ML for Life and Material Science: From Theory to Industry Applications.

[7] Qiu, K., Song, Y., Yu, J., Ma, H., Cao, Z., Zhang, Z., ... & Ma, W. Y. Unlocking the Power of Gradient Guidance for Structure-Based Molecule Optimization.

**Questions:**

1. The paper does not report structure validity evaluations (e.g., PB-valid, PoseBuster). How do the generated poses compare in physical plausibility?
2. The reward formulations in Equation 11 and Equation 50 differ. Which one is actually used to define the winner–loser pairs?
3. Given the relatively small improvements in chemical property metrics (QED, SA, Lipinski), what is the rationale for the chosen weight configuration in the reward function?
4. How sensitive is model performance to the reward weighting scheme? Have ablations been conducted?
5. Why are only 3,000 proteins sampled for generation? This limited data usage may lead to sub-optimal fine-tuning and weak generalization.
6. Were experiments repeated with multiple random seeds to ensure the statistical significance of the reported gains?

---

### Official Review · Reviewer_WWB4 · 2025-11-03

**Soundness:** 2
**Presentation:** 3
**Contribution:** 1
**Rating:** 2
**Confidence:** 4

**Summary:**

This paper addresses the challenge of scarce pocket-ligand data in Structure-Based Drug Design (SBDD) using diffusion models, proposing a novel framework called Reward-Focused Fine-Tuning (RFFT) to enhance the generation of drug-like molecules. The core idea of RFFT is to leverage self-generated samples from a pretrained pocket-aware diffusion model (e.g., TargetDiff) and a composite reward score (combining chemical properties like QED, SA, Lipinski, and binding affinity via Vina score) to construct "winner-loser" ligand pairs. These pairs serve as preference feedback for fine-tuning the model through Direct Preference Optimization (DPO), forming an iterative closed-loop to continuously improve performance.

In experiments, the authors fine-tune the pretrained TargetDiff model using RFFT (yielding TargetDiff-RFFT) and evaluate it across three key dimensions: chemical properties, protein-ligand binding affinity, and substructure fidelity. Results show that TargetDiff-RFFT achieves competitive performance in chemical property analysis (e.g., 97.41% validity and top QED score) and ranks second in binding affinity analysis (behind only MolCraft). Notably, substructure analysis reveals that RFFT not only preserves but enhances the model’s alignment with real molecular data distributions. The authors also demonstrate that RFFT’s iterative fine-tuning outperforms DPO directly applied to larger original datasets, even with fewer self-generated samples.

**Strengths:**

1. RFFT avoids over-reliance on scarce external preference data (e.g., human-annotated ligand pairs) by using the target model’s own generated ligands to construct winner-loser pairs. This closed-loop design enables the model to systematically address its own weaknesses in ligand generation, leading to continuous performance improvements over multiple fine-tuning rounds (as shown in Table 3, where 4-round RFFT outperforms 1-round and dataset-based DPO).
2. A surprising and valuable finding is that RFFT improves the model’s fidelity to real molecular distributions (via JSD and MAE metrics for atom types, ring types, and functional groups). Unlike many fine-tuning methods that sacrifice distribution alignment for performance gains, RFFT demonstrates that preference-based optimization can simultaneously boost generation quality and data fidelity, validating the rationality of its feedback design.

**Weaknesses:**

1. The core idea of using preference pairs for diffusion model optimization in SBDD is not fully novel, as it overlaps with the earlier work DecompDPO (Cheng et al., 2024, arXiv:2407.13981). DecompDPO already proposed aligning diffusion models for SBDD using multi-granularity preference pairs (at molecule and substructure levels) and integrated physics-informed terms for conformational rationality —key elements that RFFT touches on (e.g., preference-based DPO and substructure analysis) but fails to cite or discuss. This omission weakens the paper’s positioning of RFFT as a "novel framework" and prevents readers from understanding how RFFT advances beyond existing preference-based SBDD methods

2. The experimental benchmarking in Table 1 is insufficiently rigorous. The authors use "average rank" as a key comparison metric but omit critical baselines identified in literature such as REINVENT4 (a sample-efficient RL-based molecular design model). Without these baselines, the average rank metric lacks statistical power—readers cannot confirm whether TargetDiff-RFFT’s top-tier performance is due to its own merits or the limited scope of compared models. This makes the claim of "highly competitive to SOTA baselines" less convincing.

3. RFFT relies on a linear weighted reward function (R = 0.1×(QED+SA+Lipinski)+0.7×(-Vina score)) to rank ligands. As the authors acknowledge in the conclusion, this linear mapping fails to capture complex nonlinear structure-activity relationships between molecular properties (e.g., non-trivial trade-offs between QED and SA). Unlike multi-objective optimization methods that model Pareto frontiers, RFFT’s linear reward may oversimplify real-world SBDD requirements and limit its ability to generate molecules that balance conflicting properties.

**Questions:**

Please see weaknesses.

---

### Note · Authors · 2025-11-20

**Comment:**

We thank the reviewers for their time and valuable feedback. After careful consideration, we have decided to withdraw our paper and will work on further improvements.

We appreciate the opportunity to submit to ICLR and thank the program committee for their efforts.

**Withdrawal Confirmation:**

I have read and agree with the venue's withdrawal policy on behalf of myself and my co-authors.